# Learning Internal Dense But External Sparse Structures of Deep Neural Network

## Abstract

Recent years have witnessed two seemingly opposite developments of deep convolutional neural networks (CNNs). On the one hand, increasing the density of CNNs (e.g. by adding cross-layer connections) achieves higher accuracy. On the other hand, creating sparsity structures (e.g. through regularization and pruning methods) enjoys a more slim network structure. In this paper, we bridge these two by proposing a new network structure with locally dense yet externally sparse connections. This new structure uses dense modules as basic building blocks and then sparsely connects these modules via a novel algorithm during the training process. Experimental results demonstrate that the locally dense yet externally sparse structure could obtain competitive performance on benchmark tasks (*CIFAR10, CIFAR100,and ImageNet*) while keeping the network structure slim.

## 1 Introduction

Deep Convolutional Neural Networks (e.g. He et al. (2016a); Krizhevsky et al. (2012); Simonyan & Zisserman (2014); LeCun et al. (1998)) have recently made remarkable success in visual tasks. An outstanding characteristic of state-of-the-art network structures (e.g. (He et al., 2016a; Huang et al., 2017)) is that the increasing density promising a higher accuracy. The very intuitive idea of improving a neural network at the earlier period is to enlarge the scale of the network. However, flowing researches have shown that when the straightforward network structure has reached a certain depth, neither the best test accuracy nor training accuracy will increase as the network depth increases (Huang et al., 2017). It is caused by a widely known research result called vanishing-gradients. An important observation is that by increasing network density, the gradient-vanishing could be alleviated and the network could become deeper and more accurate such as ResNet (He et al., 2016a) and DenseNet (Huang et al., 2017) have shown. The basic idea of ResNet could be summarized as: $x_l = H_l(x_{l-1}) + x_l$, which uses the residual addition of previous layers as input fractures. The basic idea of DenseNet could be summarized as: $x_l = H_l(x_0, x_1, ...x_{l-1})$ which actually provides the output of every previous layer a direct path to the current layer inside each block. These shortcuts between long-distance layers could make network deeper and more accurate by propagating loss directly, and in which way network becomes denser.

Another observation is that besides the trend of being densely connected, the sparsity of weights and network structures has been a spotlight for long. Along with the network scale increasing up to millions of parameters, a crucial research for real-world applications is aimed at reducing the network complexity. Han et al. (2015b) reveals the redundancy of weights by pruning parameters and encoding shared weights matrix. To solve the problem that traditional methods mostly prune parameters weight value-wise, the follow-up papers concentrate more on channel pruning (He et al., 2017) and structure sparsity (Wen et al., 2016).

Under the inspiration of bridging these two trends and search more efficient network structures, our paper explores methods which directly introduce sparsity into network structure thus avoid pruning-after-training strategy. In neural science, papers (e.g. (Betzel et al., 2017; Sztarker & Tomsic, 2011; Gazzaniga, 1988)) concentrating on the brain structure reveal that neuron connections in brain perform a locally dense but externally sparse property as paper (Betzel et al., 2017) shown, that the closer two regions are, the denser the connections between them will be. Visual cortex papers (Belliveau et al., 1991) show that while sensory information arrives at the cortex, it is fed up through hierarchy regions from primary area V1 up to higher areas such as V2, V4 and IT. Inside of each cor-

tex layer, tightly packed pyramidal cells consist of basic locally dense structures in the brain. While our brain has been trained over time, internal densely connected modules will form a few long distance and cross-level connections to transfer information to higher hierarchy. Modular structures have shown vital importance in our brain behaviors such as specializing in information processing (Espinosa-Soto & Wagner, 2010), performing focal functions (Baldassano & Bassett, 2016), and supporting complex neural dynamics.

In this case, instead of creating local density by pruning redundancy on the trained model, we perform local density by prefixing untrained dense modules as tightly packed neuron cell in the human brain and let it evolving both the weights of itself and the sparse connection between them via training. Since DenseNet has reached theoretical densest connection status, we use a similar dense block structure with growth rate $k$, but only with very narrow channels in each block. The growth rate k (Huang et al. (2017)) is a hyper parameter in Densely Connected structures, which denotes growth rate of the input feature map scale when network goes deeper. Previous methods constructing neural modules with structural sparsity (e.g. (Xie et al., 2018; Szegedy et al., 2017)) are mostly empirically constructing the sparse connection between modules. To give more convincing guidance of forming sparse connections, we design a genetic training strategy to search an optimized connection matrix. This algorithm treats the connection matrix as the gene, and only reserves mutated individual with the best performance among others. Actually, this strategy consistently changes the input feature groups during training process, and by always counting new feature distribution in, this strategy could take similar effect as drop-out methods, thus make the model robust. Moreover, besides merely creating parallel connections between modules, our algorithm could create long-distance connections between input module and output module by a transit layer.

The experiment results demonstrate that evolving locally dense but externally sparse connections could maintain competitive performance on benchmark image datasets while using compared slim network structures. By comparison experiments, we reveal contribution proportion on the final performance of each specific connection, and by that give the principle of design sparse connections between dense modules. The main contribution of this paper is as follows:

- We enhance the hierarchical structure by utilizing the property of locally dense but externally sparse connections.
- Instead of empirically constructing module connections, we design an evolving training algorithm to search optimized connection matrix.
- We let each module choose output flow globally rather than simply creating parallel streams between modules so that the feature could flow to final layer through various depth.
- We give a detailed analysis of how different sparse connections and different module properties will contribute to the final performance. Moreover, We reveal contribution proportion on the final performance of each connection and each module by several contrast experiments, and by that give principle of design sparse connections between dense modules.

## 2 RELATED WORK

**Network architectures are becomming denser**. The exploration of network architectures has been an important foundation for all Deep Learning tasks. At the early period of deep learning, increasing the depth of a network might promise a good result as the network structure varied from LeNet to VGG (e.g. (LeCun et al., 1998; Krizhevsky et al., 2012; Simonyan & Zisserman, 2014)). Since people realize that the increasing depth of the network amplifies problem such as over-fitting and gradient-vanishing (Bengio et al., 1994), parallel structures (e.g. (Zagoruyko & Komodakis, 2016) (Szegedy et al., 2017)) and densely connected layers (Huang et al., 2017) have been introduced to increase network capacity. As DenseNet reaches the densest connection method inside each dense block, we refer to the dense block in this paper while constructing internal densely connected modules. Although our paper does not merely concentrate on highest benchmark accuracy, but also hierarchy structure and global sparsity, we still acquire competitive result on benchmark datasets using slim network structure.

**Deep neural network compression**. Besides increasing model capacity, deep neural network compression is another activate domain concentrating on acquire slim model by eliminating network redundancy. These methods could be roughly summarized as three basic aspects as fol-

lows: **1. Numerical approximation of kernels**, which includes binarization (Courbariaux et al., 2016; Rastegari et al., 2016), quantization (Zhou et al., 2017), weight sharing or coding method (Han et al., 2015a) and mainly use numerical method to approximate kernel with smaller scale; **2. Sparse regularization on kernels**, which mainly prune connections based on regularization on kernels, such as weights/channel pruning (He et al., 2017; Park et al., 2016) and structure sparsity learning (Mao et al., 2017; Wen et al., 2016); **3. Decomposition of kernels**, which mainly use smaller groups of low-rank kernel instead of a larger whole kernel, such as (Denton et al., 2014; Kim et al., 2015; Chollet, 2017) and (Xie et al., 2018). These papers mostly put an emphasis on model sparsity rather than capacity. Our paper combines the global sparsity and locally dense feature together to maintain high capacity while making the network structure slim and separable.

**Evloving algorithm on nerual network.** Many early works have developed methods that evolve both topologies and weights (Angeline & Pollack, 1993; Braun & Weisbrod, 1993; Pujol & Poli, 1998; Stanley & Miikkulainen, 2002). Most of them implement in the area of reinforcement learning. Evolutionary methods for searching better network structures have risen again recently on reinforcement domain (Togelius et al., 2009; Salimans et al., 2017). Also, it still shows great potential for image classification (Real et al., 2018). Google has proposed a state-of-the-art deep neural network structure NasNet (Zoph et al., 2017), and reaches the best performance so far by searching the best architecture on large scale. However, the huge scale of these networks with the searching parameters method still remains a problem. Our paper emphasizes on structural density & sparsity. The evolving algorithm is only used to search sparse connections during the training process.

## 3 METHODOLOGY

### 3.1 CREATE LOCALLY DENSE EXTERNALLY SPARSE PROPERTIES

Convolution operation could be understood as the projection between different channels of feature maps. Channel-wise mapping between input and output feature map could be expressed as connections. In convolution operation, the kernel could be written as: $j*i*m*n$, where $j, i$ denotes output channels and input channels, $M*N$ denotes the size of filter $W$. In order to separately represents the connection between each channel pair $(i, j)$ and illustrate concepts of 'local', we use Frobenius norm representation in Eq. (1) of each filter to represent the importance of channels as in Fig. 1:

$$F_{j,i} = (\|W\|_{\mathcal{F}})_{j,i} = \sqrt{\sum_{m=1}^{M} \sum_{n=1}^{N} w_{j,i,m,n}^2} \tag{1}$$

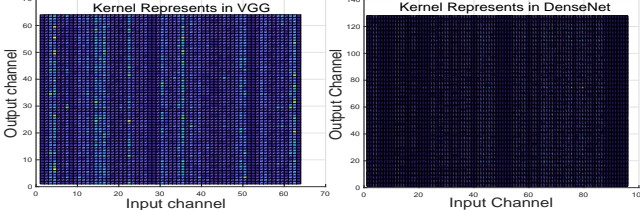

Figure 1: Separately represents connections between output channels and input channels. Brightened part means the F-norm between these specific channels is significantly large. Dark area shows that the model has significant channel wise redundancy.

Under this representation, we could calculate feature map of typical convolution kernel and show in Fig. 1. As a convolution kernel could be considered as mapping input feature from $i$ channels to $j$ channels, dark parts suggest that norm of a size $m*n$ filter is compared small, which is also called redundancy in network compression domain. Besides, inspired by the brain structure shown in neural science papers, making kernels locally dense connected could significantly save parameters. In this case, the kernel could be decomposed as it shows in Fig. 2. Obviously, this decomposition method sacrifice a large number of connections in each layer. In order to maintain high model capacity after decomposition, we create sparse connections between these modules as below.

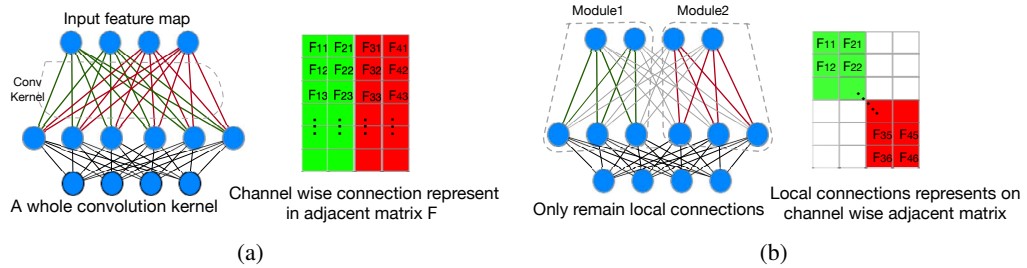

Figure 2: An example of decomposed a convolution kernel to locally dense modules where each node denotes a filter of shape $m*n$. As an example, Fig. (a) illustrates a convolution kernel with shape $6*4*m*n$ before decomposition, Fig. (b) illustrates two small kernels with shape $3*2*m*n$ after ideal decomposition. Especially, grey color denotes the connections between channels has been cut off. Note that under this example, decomposition saved $18*m*n$ parameters.

## 3.2 BUILD INTERNAL DENSELY CONNECTED MODULES

To create locally density, different from the traditional method that eliminates redundancy by pruning channels on a pretrained kernel, we would like the modularity forming along training process perusing. In that case, there exist two major ways to form local density, the first is placing L2 regularization in loss function to regulate weights distributed along diagonal in the adjacent matrix, the second and what we have chosen is to prefix some densely connected modules and explore the sparse connections between them.

In order to acquire locally density both in depth and width wise, we stack several 'narrow' convolution kernels into a dense module as shown in Fig. 3. This structure also uses a *bottleneck layer* with growth rate $k$ (Huang et al., 2017) which consists of sequential layers {BN - 1*1conv -BN - 3*3conv} (with zero padding 1). The connection strategy between bottleneck layers is also densely connected, and the connectivity could be presented as $x_l = H_l(x_0, x_1, x_2, ...x_n)$, where $H_l$ represents nonlinear operation on feature map in layer $l$ and $(x_0, x_1, x_2, ...x_n)$ represents the concatenation of all previous layers. It should be noticed that inside each dense module, the feature map size is constant, but channels will grow rapidly as layer depth and growth rate increase. To control the model scale, we use a transit layer introduced in DenseNet (Huang et al., 2017) to reduce channels of output feature map to half of original number. In this paper, we take a densely connected module as a basic element.

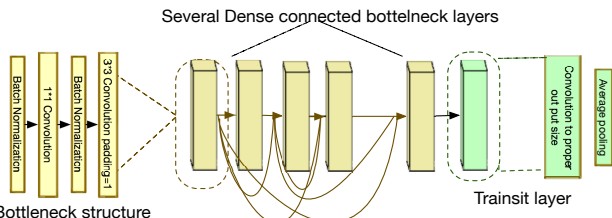

Figure 3: A structure example for a prefix dense module as shown above, where yellow layer represents several densely connect bottleneck layers (it means all output has a direct connection to the output layer). The detailed structure used in a bottleneck layer shown left. After the final layer, the green layer represents a transition layer to control the feature map size. Dense blocks depth in our experiment usually varied from 6-20 layers.

## 3.3 EXPLORE EXTERNAL SPARSE CONNECTIONS

While dealing with sparse connections between modules, a significant problem for us is that feature map size will decrease after it flows through a dense module. In order to create sparse connections,

here, we firstly figure out connection methods within the same distance, secondly we figure out methods of dealing long distance connections; finally, we could represent sparse connections in the matrix. For representing modules and its spatial relation clearly, we define a module as $M_{d,w}$, where $d$ denotes the module index of depth, $w$ denotes the module index among all of the others in the same depth.

**Connection Methods in Same Distance: Concatenation *vs* Addition**: Sparse connections might have multiple channel numbers and feature map sizes. As sparse connections might meet different channel numbers problem, there are mainly two ways for dealing it: *concatenation* and *addition*. The *concatenation method* firstly down-samples feature map size to module required size and concatenate channels as input of dense modules. The math process is defined as $o_{d,w} = M_{d,w}(o_{m_1,n_1}, o_{m_2,n_2}...)$, where $o_{d,w}$ denotes the output feature map of module $M_{d,w}$. *Addition method* actually down-samples both feature map and channel number to the required size then implies addition operation on feature maps. Similarly, the math process could be defined as $o_{d,w} = M_{d,w}(o_{m_1,n_1} + o_{m_2,n_2}...)$, where $o_{d,w}$ denotes the output feature map of module $M_{d,w}$.

To figure out the influence of the connection method, we implement a contrast experiment as it shows in *Experiment of connection method* part 4.1. Experiment results demonstrate that although concatenation method caused a larger model, its accuracy on *CIFAR10* does not show an absolute advantage over the addition method. Moreover, since we attempt to apply an evolution method to find optimized external connections, the concatenating need more operations when changing the input features. In that case, we select the addition method for sparse connections.

**Long Distance Connections**: As we mentioned above, the feature map size will change since it flows through dense modules. In order to make it possible for making long-distance connections between different depth, we use a transfer layer with {1*1conv - average pooling} structure to fit the feature map into the dense module requirement. Notice that {1*1conv} layer reform the feature map channels while average pooling changes the feature map size to fit requirement. It should be noticed that, in this way, for each module, they could have various network depth.

**Represent sparse connection**: For better analysis of sparse connections, we use the adjacent matrix to represent connections as Fig. 4. If there exists a connection, we set element value correspond to that index in connection matrix to be 1, otherwise 0. Here we could simply define the density as $D_i = \frac{\text{sum}(C_i)}{\text{sum}(C_{max})}$, where $C_i$ denotes the current connection matrix, $C_{max}$ denotes the connection matrix under the fully connected condition, sum() means the summation value of all elements in the matrix. In this paper, we only used directed graphs and down sampling connections, so the lower left of the matrix should always be zero.

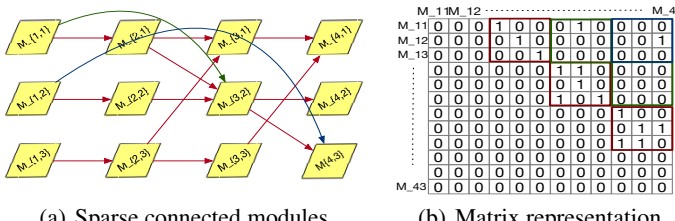

(a) Sparse connected modules          (b) Matrix representation

Figure 4: An example of using the adjacent matrix to represent sparse connections Fig. (a) in an adjacent matrix. As it shows in Fig. (b), red rectangle area denotes connections with distance 1, green rectangle denotes connections with distance 2, blue area denotes connections with distance 3

### 3.4 EVOLUTION ALGORITHM TO SEARCH IMPORTANT CONNECTIONS

One crucial problem in creating sparse topology connections is that there has not been a convincing theory on what could be called an efficient connection. In that case, we decide to make the neural network searching optimized sparse connection by itself. In this paper we use a genetic algorithm (Srinivas & Patnaik (1994)) to search the proper connections. We encoding connection matrix as the gene for genetic algorithm. In each iteration, the genetic algorithm generate several new 'individuals' with genes from mutation of the best 'individual' in last iteration. The set of generated 'individuals' is called 'population'. Genetic algorithm evolves by select best performance individual

**Encoding**: Inspired by the genetic algorithm, evolving methods need to have a good encoding to describe object features. Here we take the adjacent matrix to represent connection topology during training. In implementation details, we use a connection list to attach each module to avoid wasting storage.

**Initial state**: As we do not use pre-trained modules, we randomly initialize the weight value of modules at the first iteration of the training process. Since a deep neural network needs a long time to train, restricted to our computation capacity, we set the population between 2 to 3 individuals. For the connection matrix of the initial state, we set it only have parallel direct connections as shown in Fig. 5-*Initial State*.

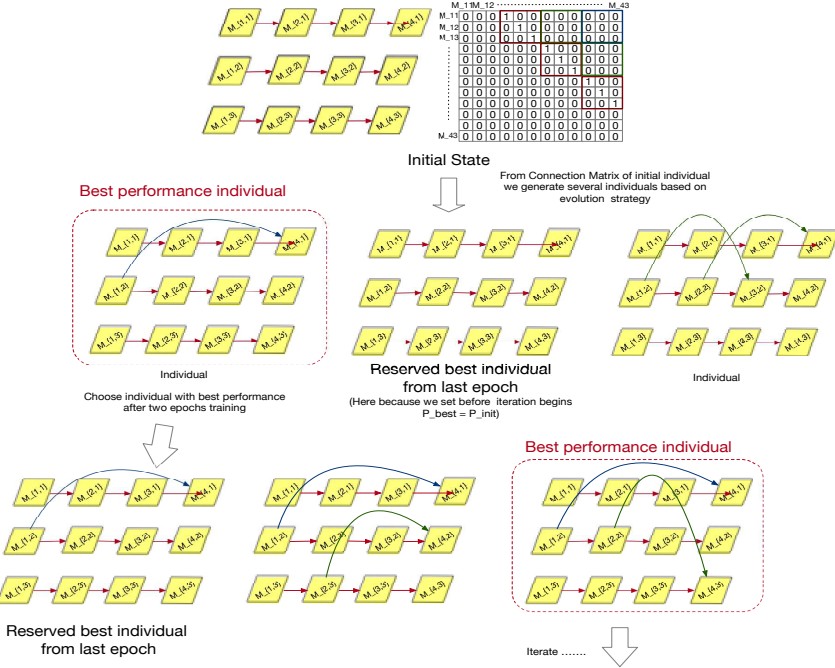

Figure 5: An example of the Network Structure Evolving. *Initial state* denotes the initial connections $P$. As we set before first iteration $P_{best} = P_{init}$, based on $P_{best}$ we generate 2 individual below. All together these 3 individual form the population to be trained simultaneously in iteration 1. Then, we choose the individual with the best performance, and based on that we form population for iteration 2. Follow this principle we maintain network evolving.

**Evolution Strategy**: We define the connection matrix of the initial individual state as $P_{init}$; best performance individual of the previous iteration as $P_{best}$, and others as $P_i$ at beginning of each iteration, the evolution of connections could be defined as:

$$P_1, P_2... = G(P_{best}) \tag{2}$$

where we choose $P_{best}$ as input of mutation function $G$, then generate several mutation individuals $P_1, P_2...$ based on $P_{best}$. Then we treat the set of $P_{best}, P_1, P_2...$ as population in this iteration. It means the best performance individual will remain to next iteration, and based on it we mutate new individuals. What exactly mutation function $G$ does is that based on the input connection matrix, randomly pick two possible connections and change the connectivity of it. It means that, if we randomly pick an unconnected connection, we set it connected, and for already connected connection, we set it disconnected. Different from methods used in the NEAT algorithm (Stanley & Miikkulainen, 2002) which forces connections denser over time, our strategy has a larger probability to become denser if density is less than 0.5, and it has a larger probability to become sparser if density is large than 0.5.

After the population of each iteration has been generated, we need to separately train each individual for a complete epoch and make it a fair comparison between each individual. In implementing detailwise, before start training, we set a checkpoint for all status and parameters and make sure that all

individuals under comparison start from checkpoints. After the training process, only the individual with the best performance will remain, and based on that, we can generate the population of the next iteration. The whole process shows in Algorithm 1 and Fig. 5.

---

**Algorithm 1** Evolutionary Connectivity Algorithm

---

1: **procedure** EVOLVE($M_{i,j}, Data, n$) ▷ $Data$:training data, $M_{i,j}$:Given Dense Modules, $n$:Total Iteration
2:     $P_{init} \leftarrow$ Initial Connection Matrix
3:     $P_{best} \leftarrow P_{init}$
4:     **for** $n$ iterations **do**
5:         $P_1, P_2...P_{k-1}, P_k \leftarrow G(P_{best})$   ▷ $k$,Number of individuals in a generation, $P_k = P_{best}$
6:         $checkpoint \leftarrow$ Model at $P_{best}$
7:         **for** $k$ iterations **do**
8:             $ResumeCheckpoint$
9:             train $P_k$
10:             **if** $P_k.accuracy > P_{best}.accuracy$ **then**
11:                 $P_{best} \leftarrow P_k$
12:             **end if**
13:         **end for**
14:     **end for**
15:     Return $P_{best}$
16: **end procedure**

---

## 4 EXPERIMENTS

### 4.1 EXPERIMENT OF CONNECTION METHOD

We firstly do a contrast experiment on *Concatenation vs. Addition* method to figure out which connection method we will use. As the test object is the connection method, we prefix a group of sparse connections and control all other training strategy and environment exactly the same, then separately train the network on the CIFAR10 dataset. We run our experiments on NVIDIA K80 GPU device. The test result is shown as Fig. 6.

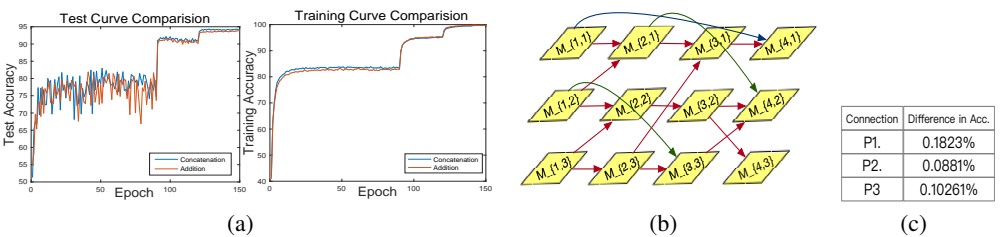

(a)            (b)            (c)

Figure 6: Comparison of Concatenation and addition method for connection. Fig. (b) denotes an example of a random chosen $P_1$ and Fig (a) denotes the train&test curve correspond to it. Fig. (c) shows the comparison result on three random chosen situation.

We could observe that the addition method only have a negligible difference with the concatenation method. Although the curve of addition method seems to have more fluctuations, it only has a negligible difference (we use the difference in highest accuracy on the test set to represent difference) with the concatenation method. As we mentioned before, the addition method is faster and more convenient for changeable feature map size. We choose addition method in later experiments. It should be also noticed that, the accuracy step jumps in the figures are caused by learning rate change for all experiments in this section. As we use the same learning rate change strategy mentioned in section 4.2 for all experiments, all step jumps in our experiments happen at the same position.

## 4.2 SPARSE CONNECTION EVOLVING METHOD

For prefixed dense modules, we set it with 4 different depth, where each depth has 3 modules. The total of 12 modules has the growth rate of 32, the modules in depth 1,2,3,4 respectively have 6,12,24,16 layers. Then we run several sparse connection evolving algorithms also training on CI-FAR10 dataset on NVIDIA AWS P3.x2large instance. We set the total iteration number to be 160, with weight decay of 5e-4. We use SDG with momentum 0.9 for gradient decsent. The learning rate strategy is the same as most of the papers that during epoch 0-90 the learning rate is 0.1; during 90-140 learning rate is 0.01; and during 140-160 learning rate is 0.001. It should be noticed that changing the learning rate will lead to accuracy 'step jumps' such as Fig. 6-9 shows. It's a common phenomenon. Restricted to our computation power, we set the number of individuals generated in each iteration to be 2. The training curve of $P_{best}$ shown as Fig 7.

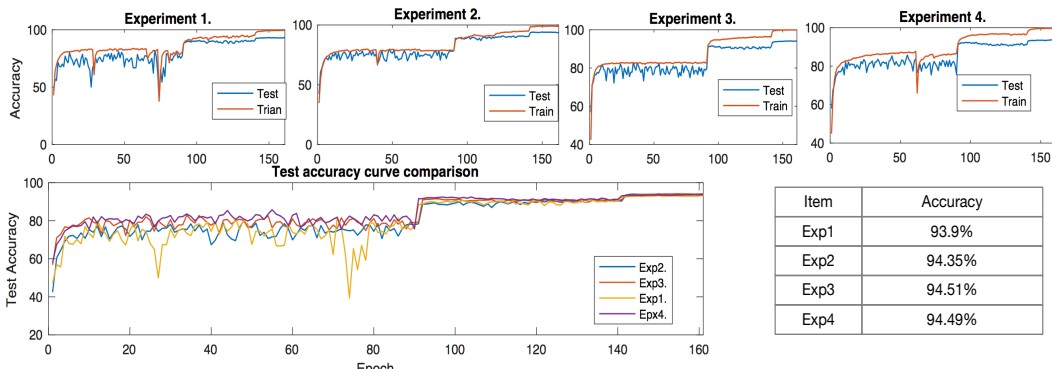

Figure 7: Several Repeatable Experiment on Sparse Connection Evolving. The upper three figures denote the training curve & testing curve of each experiment. The lower figure denotes the comparison of test accuracy of each experiment. All accuracy step jumps are caused by learning rate change strategy in section 4.2.

According to the repeatable experiments, we could see that although randomness of forming the first generation of populations may lead to variation and fluctuation in the early period of the testing performance curve, the training curve will finally converge to the same trend. This shows the *feasibility* of our algorithm. Based on these experiments we found that the optimized connection matrix is not unique to achieve good performance. However, we could still find some similarity between those connection matrices in the experiment (Fig. 8) which could reach high accuracy. It denotes that the modules with shallow depth are more likely to form a long-distance connection, which means the distance between the input feature map and output are shorten under that situation. This perfectly fits a current trend observed by various other papers (Huang et al., 2017; Redmon & Farhadi, 2018; He et al., 2016a; Chollet, 2017; Szegedy et al., 2016; 2017) that skip/direct connections are important.

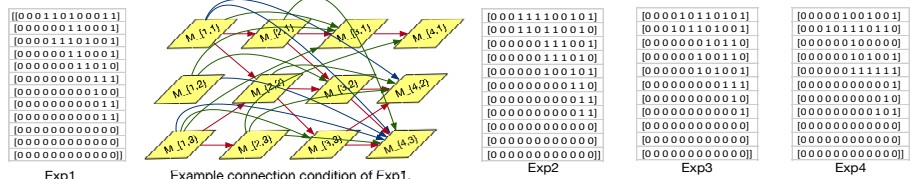

Figure 8: Connection Matrix with Best Performance on each Experiment. We also give an example connection status of Exp1.

### 4.2.1 HOW GROWTH RATE OF THE DENSE MODULE INFLUENCES THE FINAL RESULT

Here, we also make a contrast experiment by controlling all other factors the same except the growth rate $k$ in the prefix dense modules. We train the network with the same strategy and the same device

above. The result shown in Fig. 9. Clearly, the networks with smaller growth rate have higher test accuracy and more flatten curve shape compared to those with larger growth rates at the earlier period of training. It means that the modules with smaller scale are easier to train while evolving sparse connections. We can also see that although modules with smaller growth rates converge really fast and could get a good result after 90 epoch, the final test accuracy is not as high as those modules with larger growth rate. This phenomenon, in fact, proves an empirical conclusion that neural network redundancy is a necessary part of achieving high performance on test accuracy. However, experiment results also demonstrate that the network redundancy is not the 'larger the better'. As it shows in Fig. 9, after the growth rate is larger than 32, the test accuracy will not increase anymore. It is also rational because if the capacity of each module is too large, an unstable input feature may make the network harder to train. In another side, the increasing growth rate, which leads to the increasing of model scale, increases the risk of over-fitting.

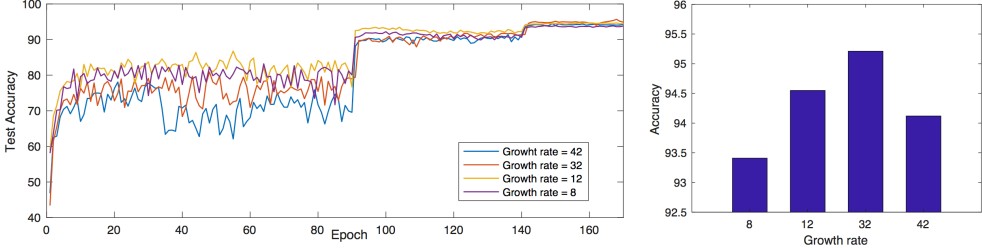

Figure 9: Test Accuracy Curve Comparison on Different Growth Rate. Each color represents test accuracy curve of experiments on different growth rate.

### 4.3 PERFORMANCE BENCHMARK

Although our paper emphasizes on how sparse connections will change the model performance, we still give performance scores on the benchmark dataset as shown in Tab1, Tab2. Since the aim of this paper is to obtain slim structures while keeping the model's capacity and achieve separable network structures, the test accuracy on both *ImageNet* and *CIFAR* is not that high compared to the state-of-the-art model. However, we still get a competitive result on both datasets.

Table 1: Test error rate performance on CIFAR dataset. Note results with * are the best result run by ourselves.

| Method | Params | Depth | CIFAR-10 | CIFAR-100 |
|---|---|---|---|---|
| Network in networkLin et al. (2013) | - | - | 8.81 | 35.68 |
| VGG19Simonyan & Zisserman (2014) | - | - | 6.58 | 27.09 |
| Highway Network Srivastava et al. (2015) | - | - | 7.72 | 32.29 |
| DFN Wang et al. (2016) | 3.9M | 50 | 6.40 | 27.61 |
| Fractral NetLarsson et al. (2016) | 38.6M | 21 | 5.22 | 23.30 |
| ResnetHe et al. (2016a) | 1.7M | 110 | 5.46 5.58* | 27.62 |
| Pre-activated ResnetHe et al. (2016b) | 1.7M | 164 | 4.72 5.12* | 25.6 |
| Wide ResnetZagoruyko & Komodakis (2016) | 7.4M | 32 | 5.4 | 23.55 |
| Densenet (k=12)Huang et al. (2017) | 1M | 40 | 5.24 5.43* | 24.42 24.98* |
| Densenet-BC (k=12) | 0.8M | 100 | 4.51 | 22.27 |
| Densenet121 (k=24) | 15.2M | 121 | 4.68* | 21.49* |
| SDMN, growth rate k=8, 6 modules | 0.4M | - | 6.97* | - |
| SDMN, growth rate k=8, 8 modules | 1.3M | - | 6.59∗ | 25.6* |
| SDMN, growth rate k=8, 12 modules | 2.3M | - | 5.97* | 24.8* |
| SDMN, growth rate k=12, 12 modules | 3.7M | - | 5.35* | 23.41* |
| SDMN, growth rate k=32, 12 modules | 22M | | 4.79* | 21.9* |

### 4.4 SEPARABLE OF SPARSE CONNECTIONS

After the evolving training algorithm gives optimal sparse connections, we wonder which sets of connections play a more important role in the whole network flow. We separately cut off one sparse

Table 2: Test accuracy rate performance on ImageNet dataset, compared with slim network models.

| Model | Params | Top1/Top5 Acc. |
|---|---|---|
| MobileNetV1 | 4.2M | 70.6% / 89.5% |
| ShuffleNet (2x) | 4.4M | 70.9% / 89.8% |
| MobileNetV2 (1.4) | 6.9M | 74.7% / |
| NASNet-A (N=4, F=44) | 5.1M | 74.0% / 91.3% |
| Sparse-Dense-Modules(k=12) | 3.7M | 71.1%/90.0% |

connection each time and test the remaining accuracy on *CIFAR10* dataset. Then we come up with a matrix that suggests how much accuracy decreasing results from losing each connection as shown in Fig 10. In experiment results, the red rectangle area denotes the direct connections; the green and blue rectangle area denote the long-distance connections. According to the accuracy loss distribution, local and direct connections are of vital importance for a neural network. It is rational because the deep learning method needs a compared invariant forward and backward feature flow path for propagation. We could also see the accuracy loss is larger along the diagonal to the high left of the matrix. It means that connections with shallow depth perform a more important role in conduct features/patterns than deeper connections. It is also rational because the shallower connections simultaneously mean the features that flow through such connections have not been extract to some level of abstraction. In Fig. 10, each column denotes how many connections are attached to this module. Contrast experiment suggests that: 1. The connections between shallow modules are more important than deeper and long-distance connections. 2. The local connections contribute a base test accuracy, and the long-distance connections will contribute more on increase accuracy by small steps based on the baseline accuracy. 3. The more connections a module has as input, the more robust the module will be when cutting off some of the connections.

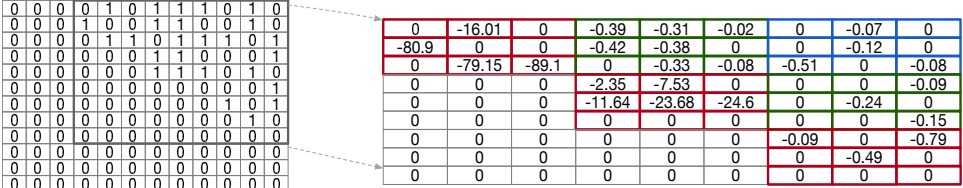

Example connection matrix                    The accuracy loss after pruning specific connection (%)

Figure 10: Example connection matrix shows the selected best connection from a typical experiment. Right part of the figure shows how much accuracy will loss if we cut off the corresponding connection in the connection matrix.

## 5 CONCLUSIONS & FUTURE WORK

In this paper, we firstly create locally dense and externally sparse structures by prefixing some dense modules and add sparse connections between them. Experiment results demonstrate that evolving sparse connections could reach competitive results on benchmark datasets. In order to give properties of these biologically plausible structures, we apply several sets of contrast experiments as shown in *Experiment*. By equally changing the input feature groups of each module during the whole training process, this strategy could alleviate the risk of the weights being trapped in local optimal point. Same to most of the related works, redundancy of each dense module is not 'the larger the better', where the test accuracy will first increase within the growth rate increases, but finally drop while the growth is above some threshold.

The combination of being dense and being sparse is an interesting area, and the internal dense and externally sparse structure also coincide with the modularity in human brain. We prove the feasibility of these structures and give a simple algorithm to search best connections. We also noticed that the connection matrix is not unique for reaching good performance. We will concentrate on revealing the relationship between these similar connection matrices and the representing features behind it. In this case, we may acquire state of the art performance on other datasets and tasks in our future

work. Moreover, as these structures have various direct paths between input and output, separating a network into several small networks without any accuracy loss is also a promising topic.

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
