# OpenReview forum: "Learning Internal Dense But External Sparse Structures of Deep Neural Network"
_ICLR.cc/2019/Conference_

### Official Review · AnonReviewer3 · 2018-11-05
**LEARNING INTERNAL DENSE BUT EXTERNAL SPARSE STRUCTURES OF DEEP NEURAL NETWORK**

**Rating:** 6
**Confidence:** 2

**Review:**

The authors bridge two components (density of CNNs and sparsity structures) by proposing a new network structure with locally dense yet externally sparse connections.

+ Combination of being dense and sparse is an interesting area.
- Although experiment results demonstrate evolving sparse connection could reach competitive results, it would be interesting to show how separating a network into several small networks is useful, for example, interpretablity of deep neural network. There is an interesting work: "Using deep learning to model the hierarchical structure and function of a cell" https://www.nature.com/articles/nmeth.4627

---

### Official Review · AnonReviewer1 · 2018-11-05
**This paper proposes a neural network architectures with locally dense and globally sparse connections. Using dense units a population-based evolutionary algorithm is used to find the sparse connections between modules.**

**Rating:** 5
**Confidence:** 3

**Review:**

The problem is of increasing practical interest and importance.

The ablation study on the contribution and effects of each constituent  part is a strong part of the experiment section and the paper.

One major concern is about the novelty of the work. There are many similar works under the umbrella of Neural Architecture search who are trying to connect different building blocks (modules) to build larger CNNs. One example that explicitly makes sparse connections between them is [1]. Other examples of very similar works are [2,3,4].

The presentation of the paper can be improved a lot. In the current setup it’s very similar to a collection of ideas and tricks and techniques combined together.

There are some typos and errors in the writing. A thorough grammatical  proofreading is necessary.

In conclusion there is a claim about tackling overfitting. It’s not well supported or discussed in the experiments.

[1] Shazeer, Noam, et al. "Outrageously large neural networks: The sparsely-gated mixture-of-experts layer." arXiv preprint arXiv:1701.06538 (2017).
[2] Xie, Lingxi, and Alan L. Yuille. "Genetic CNN." ICCV. 2017.
[3] Real, Esteban, et al. "Large-scale evolution of image classifiers." arXiv preprint arXiv:1703.01041 (2017).
[4] Liu, Hanxiao, et al. "Hierarchical representations for efficient architecture search." arXiv preprint arXiv:1711.00436 (2017).

---

> ### Author Response · Authors · 2018-11-24
> **Detailed reply about novelty and improvement of our writing.**
>
> We are very excited about the positive and enthusiastic support of our core idea. Thank you for your feedback about our strong part. We totally agree with you that our strong part is Section 4.4.
>
> About your main concerns:
> We belive we have enough novelty for our work.
>
> Paper [2] claimed that they use a genetic algorithm for searching network structures. As I understand, their work mostly concentrates on searching skip connections of layers. As it is shown in Fig. 2 in [2], the optimization object is only connections between layers, however, strictly speaking, they didn’t change the structure of the network.  Our focus is combining the locally dense and externally sparse property of the human brain into the neural network. Our optimization object is sparse connections between dense modules. In our paper, we figure out a method to achieve local density and global sparsity and demonstrate it with our solid experiments.  We have typical hierarchical structures, and our experiment figures out how different parts of the network will influence the final result. Yes, many papers could use genetic algorithms, but they all have their own contributions. Moreover, according to the experiment part in page 8 of [2], we acquire more solid experiment results.   As these two papers have different core ideas, we believe that our paper have enough novelty.
>
> Paper [3] focuses on minimizing human participation as much as possible. They search all parameters including learning rate, identity, reset weights, insert & remove convolutions.  We think paper [3] has the same motivation and idea as paper [2] that reduce human participation as much as possible. We think paper [3] is even better than paper [2] as they are in the same direction.
> Our motivation is different from these two papers.  Our focus is combining locally dense and globally sparse properties of network structures.  We do analysis about how different parts of the network or the different types of connections will influence the final performance in Section 4.4.
>
>
> Paper [4] has a similar idea of hierarchical structures as our paper. But our basic elements are modules which contain several dense layers. We notice that in their paper, evolving algorithm could form cliques in the end. We think it might have some interesting conclusions if they look into properties like density and which connections are important. We think searching network structure is a big topic. It worth many good papers on this topic. But all of them have different contributions.
> Different from their work, we focus on the implement of human-like locally dense but externally sparse structures in our paper. And we make a detailed analysis of how each long-distance connection will influence the final result.
>
> Paper [1] is a good NLP paper with special layers and searching strategy. This paper is also under the network search topic. But we focus on totally different aspects.
>
> Thank you for mentioning some typos and grammar mistakes. We apologize for this. We spend a lot of time doing several rounds of proofreading and revising. We hope this version may make you feel better.
>
> In all, although there are some papers having similar topics to our paper (network searching, hierarchical network structures, network pruning), we think a good topic worth many good papers to contribute to it. Also, we think we have enough novelty as present above. In that case, we think it worth to be accepted. Thank you very much.

---

### Official Review · AnonReviewer2 · 2018-11-10
**Interesting topic and insights, but requiring more improvements**

**Rating:** 5
**Confidence:** 3

**Review:**

The authors present the interesting and important direction in searching better network architectures using the genetic algorithm. Performance on the benchmark datasets seems solid. Moreover, the learned insights described in Section 4.4 would be very helpful for many researchers.

However, the overall paper needs to be polished more. There are two many typos and errors that imply that the manuscript is not carefully polished. Explanations about some terms like growth rate, population, etc. are necessary for broader audience.

More importantly, while some of step jumps in Figure 6~9 are suspicious, it turns out that all the step jumps happen at the same number of steps, which are identical to the change of learning rates described in Section 4.2. Thee clear explanation about that phenomena is required.

* Details
- Please represent the blocks (e.g. 1*1conv) better. Current representation is quite confusing to read. Maybe proper spacing and different style of fonts may help.
- In Page 5, "C_{m}ax" is a typo. It should be "C_{max}".
- Regarding the C_max, does sum(C_max) represent (D * W)^2 where D is the total depth and W is the total indicies in each layer? If so, specifying it will help. Otherwise, please explain its meaning clearly.
- In Figure 4(a), it would be better if we reuse M_{d,w} notation instead of Module {d_w}.
- Please briefly explain or provide references to the terms like "growth rate", "population", and "individuals".
- Different mutations may favor different hyper-parameters. How the authors control the hyperparameters other than the number of epochs will be useful to know.
- Even though the sparse connection is enforced for some reasons, overfitting, variance, or any other benefits that slim structure can bring in has not been evaluated. They need to be presented to verify the hypothesis that the authors claim.

---

> ### Author Response · Authors · 2018-11-24
> **Detailed Reply about our paper**
>
> First of all, thank you very much for writing a detailed review of our paper!
> We are excited with the positive and enthusiastic support of our core experiments.
>
> We feel very sorry for the typos and grammar mistakes in our previous version. After a long time of proofreading and revising, we believe that the current version is much better. For instance, we have provided brief explanations with references of terms like ‘growth rate’ and ‘population’ in our current version as suggested by you.
>
> Moreover, we have added a clear explanation about step jumps in Figures 6~9 in our experiment section. Yes, these step jumps are caused by the change of learning rates. As we use same learning-rate change strategy in all experiments, the step jumps of all experiments happen at the same epochs. We originally thought it was a common phenomenon. Thanks to your advice, we have added an explanation of this phenomenon in our experiment section.
>
> As we have corrected all of the typos and errors that we can find, we believe that this paper is still worth being seen by more people.
>
> Our direction is not merely using genetic algorithm to search network structure, but also is about the internal dense and external sparse network structures.  How to combine the trend of being dense and being sparse is an interesting area. Moreover, the internal dense and external sparse structure also coincides with the modularity observed in human brain. As such, we believe this paper together with its research insights might be helpful to those who want to build hierarchical network structures. We really put a lot of work in this paper. It might not be perfect right now, but we think it’s worth being seen by more people.
>
> Problem:
>
> "Please represent the blocks (e.g. 1*1conv) better. Current representation is quite confusing to read. Maybe proper spacing and different style of fonts may help"
>
> Answer:
>
> Thank you very much for this suggestion! We have used a specific format and another font to improve the represents of blocks. For example on page 5. Section Long Distance Connections.  We’ve corrected this part and updated the paper.
>
> Problem:
>
> "In Page 5, "C_{m}ax" is a typo. It should be "C_{max}”."
>
> Answer:
>
> Thank you very much for pointing out the typo. We’ve corrected it and updated the paper accordingly.
>
> Problem:
>
>  "Regarding the C_max, does sum(C_max) represent (D * W)^2 where D is the total depth and W is the total indicies in each layer? If so, specifying it will help. Otherwise, please explain its meaning clearly."
>
> Answer:
>
> The element of this matrix only denotes whether there is a connection or not, and sum (C_max) and sum (C_i) just denote the summation of all elements’ values in the matrix.
>
> For example, if the element C_i[M_11,M_21] equals to 1, it means M_11 and M_21 are connected to each other.  Our original thought is that sum(C_i) denotes the number of the connections in C_i, so it’s just simply the summation of all elements’ value in the matrix. In this case, density D is between 0 and 1, where D_max=sum(C_max)/sum(C_max) reaches value 1.  Thank you for the advice, we’ve illustrated it more clearly and updated the paper.
>
> Problem: "In Figure 4(a), it would be better if we reuse M_{d,w} notation instead of Module {d_w}."
>
> Answer: We’ve correct this part and updated the paper for all similar images.
>
> Problem: "Please briefly explain or provide references to the terms like "growth rate", "population", and "individuals”."
>
> Answer: Thank you very much for the advice! We’ve provided references to the concepts, ‘growth rate’, ‘population’ and ‘individual’ and updated the paper.
>
> Growth rate k is a concept in the paper "Densely Connected Convolutional Neural networks". The l_{th} layer has k0 + k × (l − 1) input feature-maps according to paper [1], where k0 is the number of channels in the input layer. In that case growth rate k denotes how fast the feature maps will growth when the depth increases. Population and individual are concepts in genetic algorithm. Candidate solutions are called individuals or phenotypes. A population of individuals is called ‘population’.
>
> Problem: "Different mutations may favor different hyper-parameters. How the authors control the hyperparameters other than the number of epochs will be useful to know."
>
> Answer: We think different mutations may favor different hyper-parameters, too. But we didn’t get a discipline of how different hyper-parameters will influence the model so far. We keep the hyper parameters the same during the whole experiment section, and the hyper-parameters are presented in first paragraph of Section 4.2. Our experiments show that weight decay 5*10^-4 is better than 1* 10^-4, so we suggested 5*10^-4 in our paper.

---

> > ### Author Response · Authors · 2018-11-24
> > **Detailed Reply about our paper 2**
> >
> > Problem: "More importantly, while some of step jumps in Figure 6~9 are suspicious, it turns out that all the step jumps happen at the same number of steps, which are identical to the change of learning rates described in Section 4.2. Thee clear explanation about that phenomena is required."
> >
> > Answer: Yes, we think the step jumps are caused by the learning-rate change strategy. We originally thought this was a common phenomenon. So, we didn’t discuss this (due to the page limit). But thanks to your advice,  we have explained this phenomenon in the revised paper, and updated it on the website.
> >
> > Problem: "Even though the sparse connection is enforced for some reasons, overfitting, variance, or any other benefits that slim structure can bring in has not been evaluated. They need to be presented to verify the hypothesis that the authors claim."
> >
> > Answer: Thank you very much for pointing this out! Yes, claiming that sparse connections could alleviate over-fitting does’t seem to have experiment support as much as other claims. Yet, I think it’s mostly the problem of presentation.
> >
> > Our focus is on how internal density and global sparsity could be implemented in deep neural networks and how these sparse connections will contribute to final performance such as Section 4.4.  Actually, we did some prototype experiments on comparing anti-over-fitting ability between ResNet and our network structures, and the results supported our hypothesis. But due to the page limit, we didn’t put this part in the current paper. We agree that this part needs more strong support from experiments, so we just remove the claim that sparse connections could improve anti-over-fitting ability in conclusion and put it in future works. We think locally dense but globally sparse structure is a big topic; one paper may not be enough to explore all the aspects. We will do a solid analysis of its benefit in follow-up papers.

---

### Author Response · Authors · 2018-11-24
**Paper Updated**

Thanks to all reviewers, we take a lot of time on proofreading and update a new version. Detailed modifications are separately listed under each review.
Thank you very much.

---

### Meta-Review · Area_Chair1 · 2018-12-16
**A genetic algorithm to search locally-dense and globally-sparse neural network architectures.**

**Confidence:** 3
**Recommendation:** Reject

**Metareview:**

This paper proposes a genetic algorithm to search neural network architectures with locally dense and globally sparse connections. A population-based genetic algorithm is used to find the sparse, connections between dense module units. The local dense but global sparse architecture is an interesting idea, yet is not well studied in the current version, e.g. overfitting and connections with other similar architecture search methods. Based on reviewers’ ratings (5,5,6), the current version of paper is proposed as borderline lean reject.